# C5aR agonist enhances phagocytosis of fibrillar and non-fibrillar Aβ amyloid and preserves memory in a mouse model of familial Alzheimer's disease

Elena Panayiotou[1], Eleni Fella[2], Savanna Andreou[2], Revekka Papacharalambous[1], Petroula Gerasimou[3], Paul Costeas[3], Stella Angeli[2], Ioanna Kousiappa[4], Savvas Papacostas[2,4], Theodoros Kyriakides[1,2]*

1 Neurology Clinic A, The Cyprus Institute of Neurology and Genetics, Nicosia, Cyprus, 2 Cyprus School of Molecular Medicine, Nicosia, Cyprus, 3 Karaiskakio Foundation, Nicosia, Cyprus, 4 Neurology Clinic B, The Cyprus Institute of Neurology and Genetics, Nicosia, Cyprus

* theodore@cing.ac.cy

**Data Availability Statement:** All relevant data are within the manuscript and its Supporting Information files.

## Abstract

According to the amyloid hypothesis of Alzheimer's disease (AD) the deposition of prefibrillar and fibrillar Aβ peptide sets off the pathogenic cascades of neuroinflammation and neurodegeneration that lead to synaptic and neuronal loss resulting in cognitive decline. Various approaches to reduce amyloid load by reducing production of the Aβ peptide or enhancing amyloid clearance by primary or secondary immunization have not proven successful in clinical trials. Interfering with the normal function of secretases and suboptimal timing of Aβ peptide removal have been put forward as possible explanations. Complement, an innate component of the immune system, has been found to modulate disease pathology and in particular neuronal loss in the AD mouse model but its mechanism of action is complex. C1Q has been shown to facilitate phagocytosis of Aβ peptide but its Ablation attenuates neuroinflammation. Experiments in AD mouse models show that inhibition of complement component C5a reduces amyloid deposition and alleviates neuroinflammation. Phagocytes including microglia, monocytes and neutrophils carry C5a receptors. Here, a widely used mouse model of AD, 5XFAD, was intermittently treated with the oral C5a receptor agonist EP67 and several neuronal and neuroinflammatory markers as well as memory function were assessed. EP67 treatment enhanced phagocytosis, resulting in a significant reduction of both fibrillar and non-fibrillar Aβ, reduced astrocytosis and preserved synaptic and neuronal markers as well as memory function. Timely and phasic recruitment of the innate immune system offers a new therapeutic avenue of treating pre-symptomatic Alzheimer disease.

## Introduction

Alzheimer's disease (AD) is primarily a progressive neurodegenerative disease of the brain causing either sporadic or familial dementia. AD is by far the commonest dementia; the sixth

**Funding:** The author(s) received no specific funding for this work.

**Competing interests:** The authors E.P. and T.K. have filed a PCT application on the use of the modified C5aR agonist EP67 in cerebral and peripheral amyloidoses (PCT/EP2018/053362). This does not alter our adherence to PLOS ONE policies on sharing data and materials.

leading cause of death in the general population and the fifth leading cause of death in people over the age of 65[1].

The main hallmark of AD is the extracellular deposition of Aβplaques caused by the proteolytic cleavage of the amyloid precursor protein (APP). According to the amyloid cascade hypothesis genetic mutations in certain genes (e.g. APP and PSEN1/2) or age related Aberrations in the metabolism of APP result either in excess production or reduced clearance of Aβ, thus causing the deposition of Aβ oligomers and plaques in the brain due to "aggregation stress"[2].

Studies in AD patients and transgenic mice have indicated immune cell infiltrates in Aβ plaques. These cells consist of activated microglia and reactive astrocytes which are key players in modulating neuronal loss [3, 4]. Furthermore, there is evidence that peripheral phagocyte may ingress across the blood brain barrier which localizes to amyloid plaques and includes both macrophages and neutrophils [5–7]. The presence of these blood-borne phagocytes potentially modulates disease progression by attenuating the accumulation of Aβ plaques [8]. Considering these recent data it may be therapeutically prudent to enhance phagocytosis of Aβ by activating local microglia and also recruit phagocytes from the periphery in an attempt to interrupt the amyloid cascade.

The complement cascade has been shown to be activated from the very early stages of Alzheimer disease [9]. Complement component C5a is thought to bind C5a receptors on phagocytic cells such as microglia and potentially augment phagocytosis of Aβ and pre-fibrillar Aβ [10]. The C5a receptor is found on a variety of cells, mainly involved with phagocytosis and inflammatory responses. Most notably, C5a receptors are found on granulocytes, monocytes, dendritic cells, astrocytes and microglia. The C5a protein fragment is released from the cleavage of complement component C5 by protease C5-convertase which yields the fragments C5a and C5b. C5a is an extremely inflammatory molecule which boosts complement activation, attracts innate immune cells and even releases histamine in allergic responses. C5 originates in the hepatocytes even though it is also synthesized in macrophages and other cells where local demand for C5a increases. Therefore, C5a is both an anaphylatoxin and a chemoattractant which is essential not only in innate immune responses but is also involved with adaptive immunity [11].

EP67 is a conformationally biased, response-selective analogue of the biologically active C-terminal region of human complement component C5a 65–74 (Tyr-Ser-Phe-Lys-Asp-Met-Pro-(N-methylLeu)-D-Ala-Arg). The agonist was originally generated by substituting certain amino acids in C5a and attaching a methyl group to the nitrogen atom on the amide bond between proline and leucine. As a result, these structural adjustments restrict and extend the backbone's conformation thus creating biased topochemical features that allow for a conformational distinction between C5a-like inflammatory and immune stimulatory activity. These features therefore allow the peptide to interact with C5a receptors expressed on antigen presenting cells but is devoid of C5a-like neutrophil and neutropenic abilities [12].

In a recent study, AD mice vaccinated against C5a via C5a related peptides either early or late in the course of the disease achieved reduction of amyloid load only with early but not with late vaccination[13]. This suggests that C5a activity (perhaps via microglia activation) may be particularly detrimental in the early stages of AD.

Thus microglial activation may have a dual effect, driving neuroinflammation and exacerbating amyloidogenesis but also enhancing phagocytosis. We hypothesize that both the degree and timing of complement activation in relation to the stage of the disease may be of paramount importance. Herein we present data showing that intermittent administration of a modified C5a receptor agonist EP67 in an AD amyloidosis mouse model (5XFAD) results in

enhanced phagocytosis, of both fibrillar and non-fibrillar Aβ, to a level that normalizes behavioural performance and preserves synaptic integrity.

## Materials and methods

### Animals and tissue handling

The 5XFAD transgenic mouse model (Tg6799) previously described by Oakley et al. exhibits early and aggressive cerebral amyloid pathology and recapitulates many but not all AD hallmarks, it specifically does not present any tau accumulation [14]. Hemizygous mice were bred with B6SJLF1/J hybrids (Jackson Laboratories) in order to produce mice exhibiting the 5XFAD phenotype. These mice contain the known Familial Alzheimer's disease (FAD) mutations APP K670N/M671L (Swedish), I716V (Florida), V717I (London) and PS1 L286V and M146L. Standard PCR reactions were carried out to identify the mice expressing all 5 mutations (PCR protocol as indicated by Jackson Laboratories stock #: 006554).

All animals were kept in a regular 12-hour light-12 hour dark cycle and were given free access to water and food, under specific-pathogen-free (SPF) conditions. Animals were separated in cages depending on their age and treatment. During each treatment cycle EP67 (synthesized by Thermo, purity≥ 97%) was added to the animals' water source at 20g/ml. Groups were arranged as described in Fig 1. More specifically, 3 groups of age/sex matched animals were kept, one group of 5XFAD animals which were not treated with EP67, a group of 5XFAD animals treated with the peptide and finally a group of wild type B6SJLF1/J animals. At the beginning of the 3rd month of age the 5XFAD animals to be treated were given EP67 in their drinking water for one week.

At the end of the week, a specific number of animals from all 3 groups were sacrificed after undergoing the Y-maze test. The rest of the animals were sacrificed on the second week of the 6th month. The 5XFAD animals treated with EP67 were given EP67 in their drinking water during the first week of every month up until the 6th month of age, as indicated in the diagram.

All animal experiments were carried out in accordance to the 86/609/EEC Directive (Cyprus Veterinary Services committee examined and approved the following experimental and sacrifice protocol, project licence CY/EXP/P.L6/2010).

Mice were anesthetized and then euthanized using Tribromoethanol (Avertin–prepared by dissolving 2,2,2-tribromoethyl alcohol, Sigma-Aldrich T48402, with 2-methyl-2-butanol, Sigma-Aldrich 240486, and distilled water) though IP injection at a dose of 250 mg/Kg. Blood was then extracted via cardiac puncture and then the mice were exsanguinated through the

| | Month 1 | Month 2 | Month 3 | Month 4 | Month 5 | Month 6 |
|---|---|---|---|---|---|---|
| 14 5XFAD | | | S⁶ | | | S⁶ |
| 14 5XFAD EP67 | | D | S⁶ / D | D | D | S⁶ |
| 8 WT | | | S⁴ | | | S⁴ |

D: EP67 20μg/mL dosage for 1 week
Sⁿ: Sacrifice n animals

**Fig 1. Dosing and sacrifice schedule.** Illustration of the dosing schedule exhibiting the treatment of the 3 groups of mice kept.

same route with PBS. The animals were then decapitated and the whole brain was harvested and the two hemispheres were separated. One hemisphere was used for immunohistochemistry by carrying out overnight 4% PFA fixation followed by wax embedding and the other was kept at at -80˚C until further processing in order to be used for protein expression analysis. The hemisphere kept at -80˚C was initially pulverized and sonicated with PBS supplemented with protease inhibitor cocktail (approximately 50μL) in order to homogenize the tissue, resulting in a thick paste consistency. This tissue preparation was used as the starting material for both immunoblotting and immunoassays.

## Thioflavin-S positive plaques quantification

Thioflavin S staining combined with Aβ immunofluorescence were used to identify Aβ derived fibrillar (insoluble) amyloid deposits in paraffin sections obtained from sagittal brain sections. Sections were incubated with the Amyloid-β antibody (Santa Cruz sc-28365 1/500), visualised with an anti-mouse Alexa Fluor 555 (A-31570) secondary antibody (Fig 2i). Also, the 6E10 antibody (Covance 1/200) was also used to confirm amyloid deposits (Fig 2ii) but not for quantification. Sections were then stained with aqueous 1% Thioflavin S solution (T1892-25G) Plaques positive for both Thioflavin S and β-Amyloid were quantified using the ImageJ software set to measure yellow (570–585 nm). Amyloid plaques were quantified in the cortex and hippocampus (HPC) where a percentage of the surface area occupied by plaques was measured and an average percentage was obtained over fifteen serial sections. Pictures were taken using a Zeiss AXIOIMAGER M2 fluorescence microscope.

## Soluble Aβ quantification

Guanidinium chloride (GuHCL) soluble Aβ was quantified via ELISA measuring both Aβ40 and Aβ42 separately (Novex KMB3481 and KMB3441). 100mg from the whole hemisphere homogenate was lysed with Guanidine/Tris HCl buffer according to the manufacturer's instructions. The original samples were dilutes by x1000 using the kit provided reaction buffer. Briefly, the mouse detection antibody was initially incubated for 1 hours at room temperature and following the required washes the anti-rabbit IgG HRP was then incubated for 30 minutes

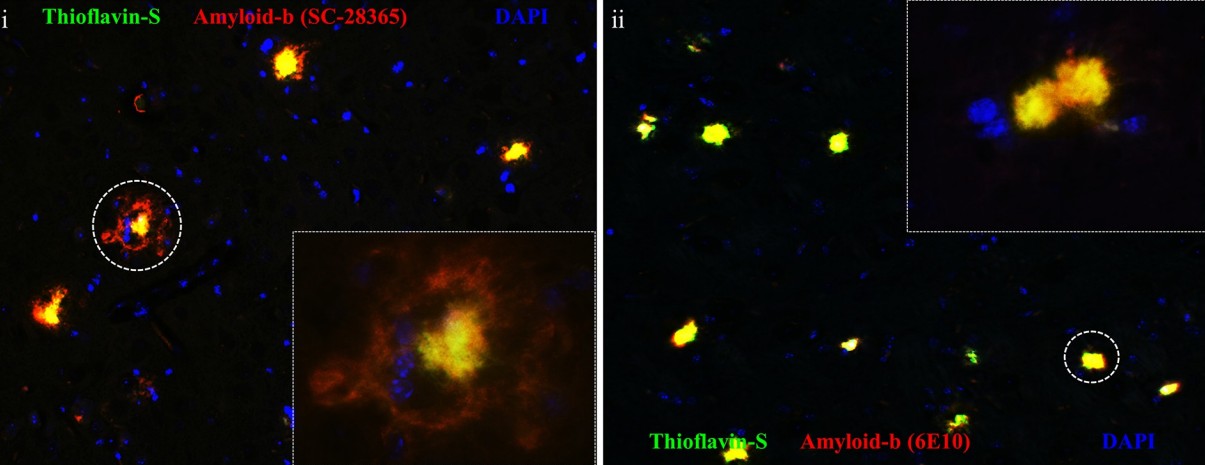

**Fig 2. Thioflavin-S positive plaque quantification.** Tissue sections were incubated with an an antibody against amyloid-β, SC-28365 (i) or 6E10 (ii) (-red- excitation λ: 555 emission λ:580). The sections were then co-stained with Thioflavin-S (-green- excitation λ: 430 emission λ:550). Areas of co-localisation were considered as fibrillar (insoluble) amyloid deposits.

again at room temperature. Again following washes the chromogen was incubated for a further 30 minutes at room temperature before halting the reaction with the stop solution. The absorbance was measured at 450 nm using a microplate reader immediately after stopping the reaction.

The hemisphere kept at -80˚C was initially pulverized and sonicated with PBS supplemented with protease inhibitor cocktail (approximately 50μL) in order to homogenize the tissue, resulting in a thick paste consistency. This tissue preparation was used as the starting material for both immunoblotting and immunoassays.

## Spontaneous alternation Y-maze

Spontaneous alternations scores were obtained for each participating mouse by carrying out the protocol as previously described [15]. Each mouse was allowed to explore the maze freely for a total of 8 minutes. During each session the sequence of entries was recorded, as well as the total number of entries. The equation used to calculate percentage alternation was: number of triads completed over the number of maximum possible alternations (total number of arms entered-2).

## Immunoblotting

Brain hemispheres were lysed in 200–300 μl RIPA buffer supplemented with protease inhibitors using sonication (10 cycles of 8–10 seconds at 60 Hz with 8–10 seconds interval time on ice). Following 30 minutes incubation on ice, homogenates were then centrifuged at 12000 x g for 15 minutes at 4˚C and the supernatant containing the proteins was collected. Protein concentration was measured using the BCA assay (Thermo Scientific, 23227) and 100 μg of proteins were mixed with 1X loading buffer supplemented with β-mercaptoethanol. Before loading into gels, sample denaturation was performed at 95˚C for 5 minutes. Protein samples were then separated via reducing SDS-PAGE in 12% or 10% separating gels and 4% stacking gels for 1 hour and 30 minutes and transferred onto PVDF membranes for 1 hour at 100 Volts. The membranes were blocked with 5% BSA in PBS-T (1X PBS with 0.1% Tween Twenty) for 1 h at room temperature and incubated overnight at 4˚C with the appropriate primary antibody in 5% BSA (PBS-T). The membranes were washed three times with PBS-T and then incubated with the appropriate secondary antibody, conjugated with HRP (diluted in 5% BSA (PBS-T)) for 1h at room temperature. The membranes were then again washed thrice with PBS-T and the specific antibodies were visualized using the Super Signal West Femto Maximum Sensitivity Substrate (Thermo Fisher 34095) as directed by the manufacturer. Antibody incubations and washes were performed with agitation.

Blots were repeated in triplicates and were visualized using the UVP bio-imaging system and the VisionWorks Software. The antibodies used for immunoblotting were against: Synaptophysin–a molecular marker for the presynaptic spines–(anti-rabbit Abcam Ab32127 1/400), GFAP–an intermediate filament protein expressed by astrocytes–(anti-mouse Sigma G3893 1/500), F4/80 –mature mouse macrophage and microglial marker–(anti-rabbit Santa Cruz sc-25830 1/200), CD88 –marker for the C5a receptor–(anti-mouse Santa Cruz sc- 53795 1/100) and LY6G –a marker for neutrophils–(anti-mouse Antibodies online ABIN361224 1/1000). The appropriate HRP conjugated secondary antibodies were used: anti-mouse (Santa Cruz SC-2031 1/5,000), 235 anti-rabbit (Santa Cruz SC-2004 1/5000).

The Image J image processing program was used to carry out densitometry calculations, while all bands were normalized against a GAPDH loading control (Santa Cruz sc- 25778 1/1000), while the same reference sample was used in all westerns to allow cross-gel comparison.

The secondary antibodies were used in 1/5000 (anti-rabbit HRP Santa Cruz sc- 2004 and anti-mouse HRP Santa Cruz sc- 2031).

## Immunohistochemistry

Paraffin sections from 4% paraformaldehyde fixed hemispheres (5μm) from sagittal brain sections were used for immunohistochemistry. Immunohistochemistry sections were used to observe staining localization on the tissue and confirm immunoblotting results. Sections were allowed to dry overnight at 55˚C and were then on blocked with 5% BSA for 1 hour. Primary antibodies diluted in 4% BSA were allowed to incubate on the sections overnight at 4˚C. The sections were then washed with PBS and the appropriate secondary antibodies diluted in 5% BSA were added for 1 hour at room temperature. Sections were again washed and dipped in diluted DAPI for 20 sections before the final wash. DAKO fluorescence mounting medium was used at the end (S3023). Pictures were taken using a Zeiss AXIOIMAGER M2 fluorescence microscope and a Leica TCSL confocal microscope.

The primary antibodies used were against: Synaptophysin–a molecular marker for the presynaptic spines–(anti-mouse Abcam SY38 1/400), NeuN–marker identifying mature neurons–(anti-rabbit Abcam EPR12763 1/600), GFAP–an intermediate filament protein expressed by astrocytes–(anti-mouse Sigma G3893 1/300), F4/80 –mature mouse macrophage and microglial marker–(anti-rat Abcam CI:A3-1 1/600), and neutrophil elastase (ELANE)–found in the azurophil granules of polymorphonuclear leukocytes–(anti-rabbit Abcam Ab68672 1/400). The appropriate Invitrogen Alexa Fluor 555 fluorescence secondary antibodies were used: anti-rabbit (A-21428 1/2000), anti-rat (A-21434 1/2000) and anti-mouse (A-31570 1/2000).

## Gene expression assay in brain tissue

Two-step RT-qPCR was carried out to quantify the expression of various phagocyte markers in the brain. 5mm brain sections from the paraffin embedded fixed hemisphere also used during immunohistochemistry were used to extract total RNA (QIAGEN RNeasy FFPE Kit 73504 used according to manufacturer's instructions). RNA concentration was assessed by Nanodrop2000 and maximum 100ng/μl was used for cDNA synthesis. The Invitrogen SuperScript™ II Reverse Transcriptase (18064–022) was used to synthesize first-strand cDNA according to the manufacturer's instructions.

TaqMan® Gene Expression Assays for mouse MCP-1 –monocyte chemoattractant protein-1 the main ligand of CCL2, which regulates both the migration and infiltration of monocyte/macrophages–, LY6C –marker denoting both macrophages and microglia–, CCR2 –chemokine receptor expressed on monocytes, microglia and T cells–and MIP-2a –chemotactic agent for polymorphonuclear leukocytes secreted by monocytes/macrophages–were then used, containing a pair of unlabelled PCR primers and a TaqMan® probe with a FAM™ dye label on the 5' end, and minor groove binder (MGB) non-fluorescent quencher (NFQ) on the 3' end. The GAPDH (4352932E) gene was used as an endogenous control in delta Ct normalization of samples. For each brain sample duplicate reactions were run.

## Statistical analyses

Statistical analysis was performed using GraphPad Prism version 5.00 for Windows (GraphPad software, San Diego California USA) where one-way ANOVA followed by Tukey's post-hoc test was carried out. Using this information, graphical charts representing the data were prepared.

## Results

### EP67 reduces both Thioflavin-S positive plaques and GuHCl soluble Aβ

EP67 is a response-selective analogue of the biologically active C-terminal region of the human complement component C5a 65–74 devoid of anaphylactoid activity [12, 16]. Therefore, the EP67 molecule, while been able to activate antigen presenting cells (APCs), lacks the ability to specifically activate neutrophils.

To evaluate the effect of EP67 on amyloid deposition in the brain, 14 (7 male and 7 female) 3 month old 5XFAD mice (previously generated and characterised by Oakley et al.,) were given 20 μg/ml of EP67 in their drinking water for one week. At the same time 14 age/sex matched 5XFAD mice and 8 wild type (B6SJLF1/J) animals were kept as controls on plain drinking water. The time point of 3 months was chosen since cerebral Aβ40 and Aβ42 levels were previously shown to increase exponentially in 5XFAD mice after the age of 2 months[14]. Six 3 month old 5XFAD EP67 treated mice were sacrificed following one-week treatment. Similarly, 6 control 5XFAD mice were also sacrificed. Considering the steep increase in cerebral Aβ in the 5XFAD mice, the remaining 8 animals were treated with EP67 for one week at the end of every month from the end of month three until they reached their 6th month of age, when they received their final treatment (a total of four cycles) tested in the maze and sacrificed. The 8 control 5XFAD mice maintained on plain drinking water were also sacrificed. During this time wild type B6SJLF1/J mice were also sacrificed at the age of 3 and 6 months (n = 4 for each age point, 2 male and 2 female).

Presence of the EP67 peptide was not analysed directly in the brains of treated animals. However animals treated with EP67 were found to have significantly higher expression levels of the C5a receptor (CD88) in the brain when compared to their wild type counterparts than untreated 5XFAD animals indicative of a likely local effect of EP67 following its administration (Fig 3). Representative sections from the cortex, thalamus and hippocampus (HPC) double stained with an antibody against amino acids 672–714 of human origin β-amyloid (suitable for detecting both β-amyloid and amyloid A4) and the amyloid plaque stain Thioflavin-S are shown in Fig 4a. The wild type mice at 3 and 6 months of age exhibit no staining for amyloid-β or Thioflavin-S as expected (Fig 4ai–4aiii and 4ax–4axii). Both the control 5XFAD and EP67 treated 5XFAD mice exhibit staining for plaques in the cortex, thalamus, HPC at the age of 3 months (Fig 4aiv, 4av, 4avi, 4avii, 4aviii and 4aix respectively) which appear more prominent in the untreated animals. At the 6-month age point the untreated control 5XFAD mice exhibit a greater load in all three regions examined (Fig 4axiii–4axv) when compared to the EP67 treated mice (Fig 4axvi–4axviii). Amyloid plaques were quantified in both the HPC and cortex through Thioflavin-S staining. Animals treated with EP67 displayed lower levels of amyloid when compared to their untreated 5XFAD counterparts (Fig 4b and 4c).

Quantification of both GuHCl soluble, Aβ40 (Fig 4d) and Aβ42 (Fig 4e) was carried out using immunoassays against these peptides. While Aβ40 levels did not appear to differ at the 3-months, significant reduction was observed in the 6 month EP67 treated animals compared to untreated. It should be noted that the 5XFAD mice exhibit Aβ accumulation by 3 months of age and as the mice age this is predominantly Aβ42 rather than Aβ40 [14]. With regards to Aβ42, a significant reduction was recorded at both the 3-month and 6-month age points in the treated animals, with the greatest reduction observed following four cycles with EP67.

EP67 appears to affect the rate of amyloid accumulation since by 6 months of age the levels of Aβ42 were nearly halved and Aβ40 was decreased by approximately 25% compared to untreated animals.

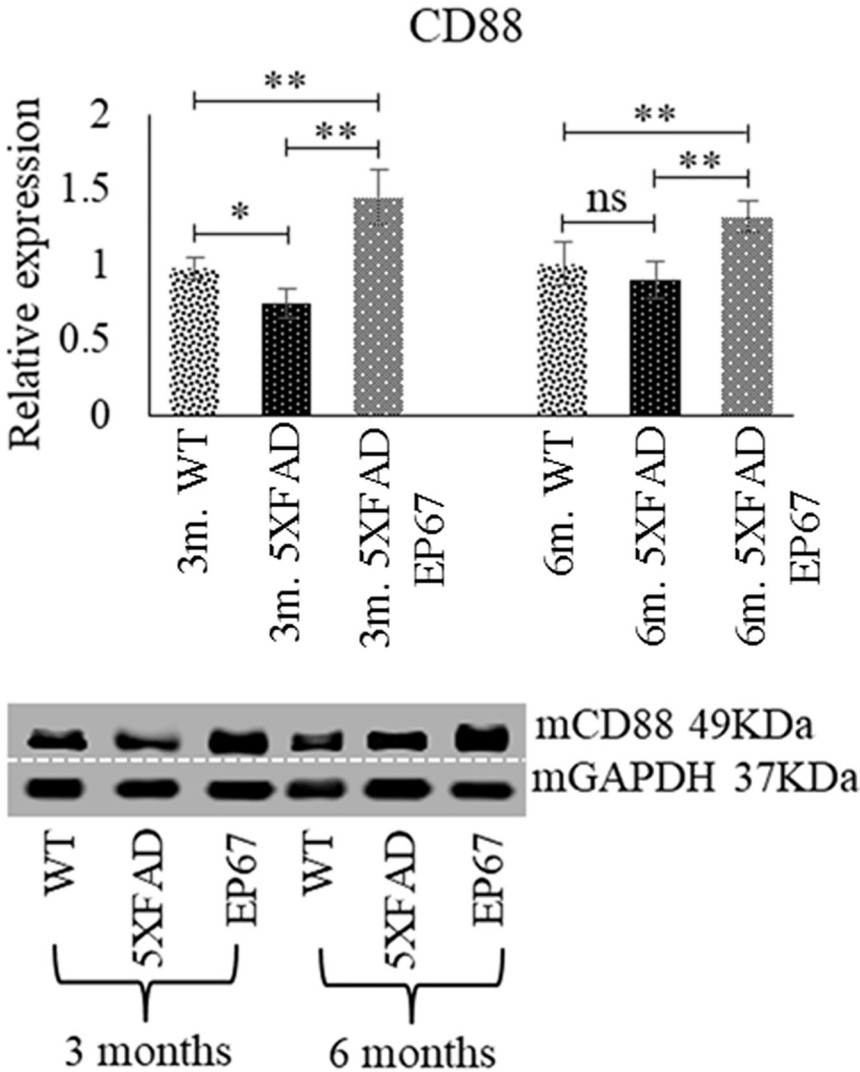

**Fig 3. C5aR (CD88) expression.** Immunoblot analysis of the CD88 marker revealing increased expression of the receptor in 5XFAD animals treated with EP67, both at 3 months and 6 months compared to both the wild type and untreated 5XFAD animals. (Immunoblot images indicate representative sample runs and were cropped as indicated by the dotted white line).

## EP67 protects against short-term spatial working memory loss

Aβ1–42 oligomers have previously been shown *in vivo* to be neurotoxic partly as a result of diminished synaptic activity in HPC [17, 18]. All mice were evaluated via the Y-maze task. The results indicate an age related cognitive decline in control 5XFAD mice when compared to wild type mice (Fig 5a). EP67 treated 5XFAD treated mice exhibit significant sparing in short-term spatial working memory when compared to their untreated control 5XFAD counterparts. In addition, the total number of arm entries was comparable for all groups of mice signifying no impairment in motor function which would have affected the mice's explorative ability (Fig 5b). Therefore, EP67 appears to protect short-term spatial HPC associated memory.

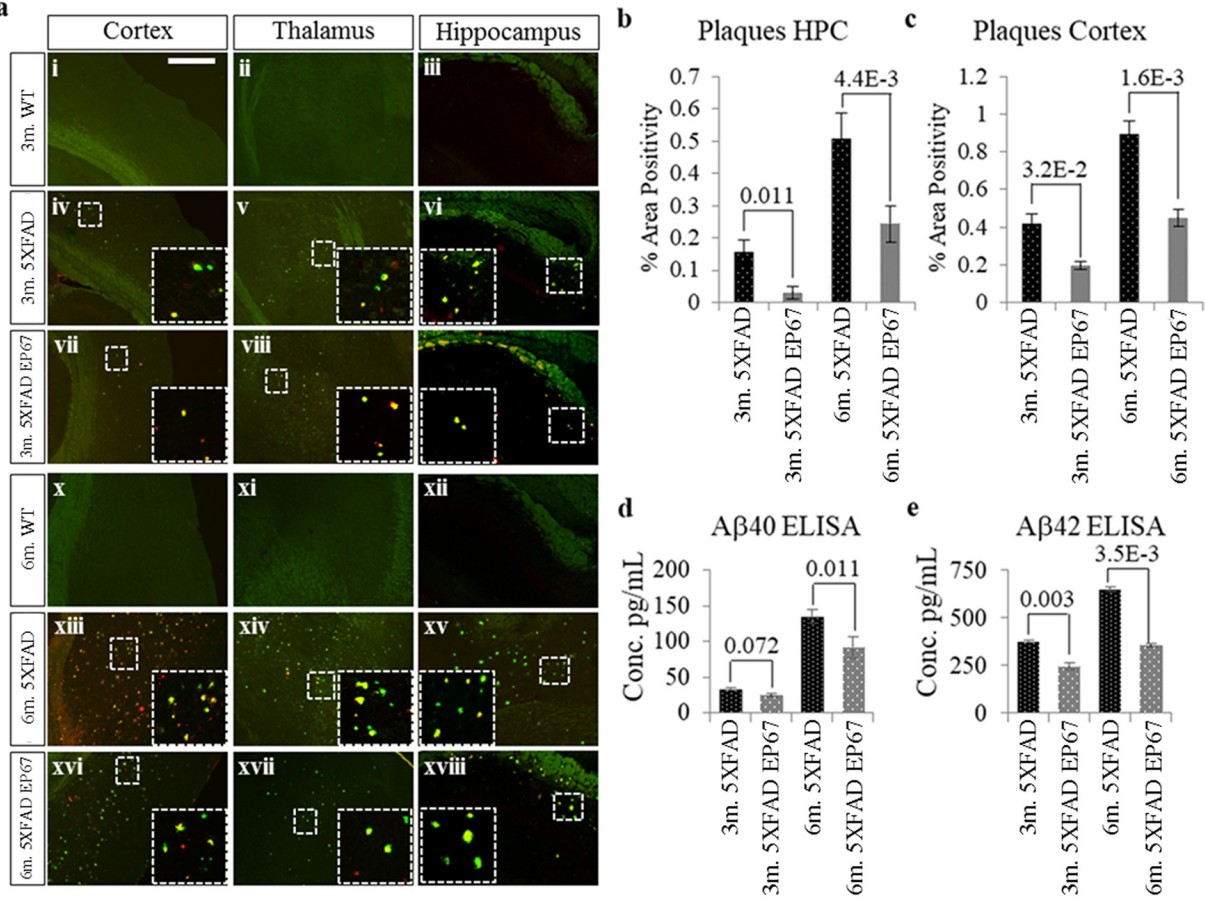

**Fig 4. Amyloid plaques deposition.** *(a)* Representative sagittal brain sections from 3 and 6 month old wild type mice (ai–aiii, x–xii respectively), 5XFAD (iv–vi, xiii–xv respectively) mice and 5XFAD mice treated with EP67 (vii–ix, xvi–xviii respectively). These sections were co-stained with Thioflavin-S (-green- excitation λ: 430 emission λ:550) and an anti-amyloid-β antibody (-red- excitation λ: 555 emission λ:580). Representative images of the cortex, thalamus and HPC are shown. Scale bar: 300μm. Serial sections were double stained with Thioflavin-S and an anti β-Amyloid antibody to quantify amyloid plaque deposition in the hippocampus (HPC) *(b)* and cortex *(c)*. n = 6/group/age. Mean ± 1SD. *(d&e)* Whole hemisphere brain homogenates from wild type, 5XFAD and 5XFAD mice treated with EP67 at ages 3 and 6 months were used to measure the amount Aβ 40 *(d)* and Aβ 42 *(e)*, n = 6/group/age.

## EP67 prevents synaptic and neuronal loss

Early accumulation of neurotoxic Aβ has been hypothesized to be one of the initial triggers leading to neurodegeneration [19]. In order to investigate synaptic loss we used antibodies against the synaptic marker synaptophysin (Fig 6).

Western blot analysis of synaptophysin revealed that the control 5XFAD mice exhibit severe decrease in synaptophysin expression at both 3 and 6 months when compared to the wild type animals, whist 5XFAD animals treated with EP67 do not. Further investigation with immunohistochemistry confirmed these findings (Fig 7). Similar results were obtained with the neuronal antibody against the post-mitotic neuronal marker NeuN (Fig 8) where a dramatic decrease in NeuN expression in the 5XFAD animals at 3 and 6 months when compared to the wild type and EP67 treated 5XFAD mice. Again EP67 treated 5XFAD mice appeared to possess similar levels of expression of the neuronal marker as the wild type mice.

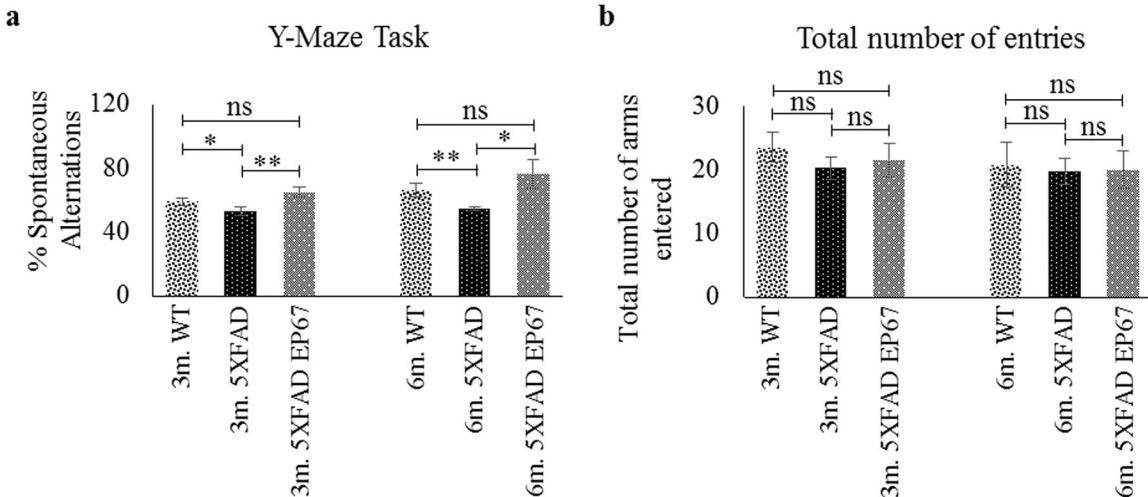

**Fig 5. Y-maze task.** *(a)* Wild type, 5XFAD and 5XFAD EP67 treated mice of 3 and 6 months of age were given the spontaneous alternation behavioural test using a Y-maze. *(b)* The number of arm entries for each group was recorded and exhibited no significant difference among any group of animals. n = 6/group/age. Mean ± 1SD.

## EP67 reduces astrocytosis

Astrocytosis has long been recognized as part of the neuroinflammation observed in both AD brains and animal models and thought to be a result of amyloid deposition[20, 21]. The astrocytic marker of glial fibrillary acidic protein (GFAP) was used to evaluate the distribution of astrocytes in the brains of EP67 treated and control 5XFAD brains (Fig 9a). Western blot analysis of 3 month old 5XFAD mice with GFAP revealed an increased expression of the marker when compared to their wild type control mice, while in EP67 treated mice GFAP expression appears to be significantly decreased when compared to the untreated 5XFAD animals. Immunohistochemistry revealed a great number of astrocytes in the 5XFAD untreated animals (Fig 10b and 10b') while very limited staining was observed in the 6 month old EP67 treated 5XFAD mice (Fig 10c and 10c'). GFAP expression in the EP67 treated 5XFAD animals did increase in the older animals as compared to their 3 month old counterparts. Both the immunohistochemical and immunoblotting analysis showed that expression of the astrocyte marker GFAP is significantly decreased following treatment with EP67.

## EP67 administration increases phagocytosis of amyloid plaques

Microgliosis is another feature of neuroinflammation. C5a receptors are carried by macrophages and neutrophils and also neurons, microglia and astrocytes within the brain [22–24]. Simard et al., demonstrated that peripheral phagocytes may be capable of migrating towards Aβ plaques, a response elicited by Aβ40 and Aβ42, in order to aid clearance of the plaques through phagocytosis [8].

We investigated phagocytosis through the mouse specific marker F4/80 (Figs 9b and 11), as well as Iba1 and CD68 (data not shown). Comparable results were obtained with all the monocyte/macrophage/microglia markers. There was a general increase in the expression of F4/80 in control 5XFAD mice at both 3 and 6 months when compared to the wild type controls; however an even greater increase was observed in the EP67 treated 5XFAD mice (Figs 9b and 11). Our observations indicate a significant increase in the presence of phagocytic cells in EP67 treated animals.

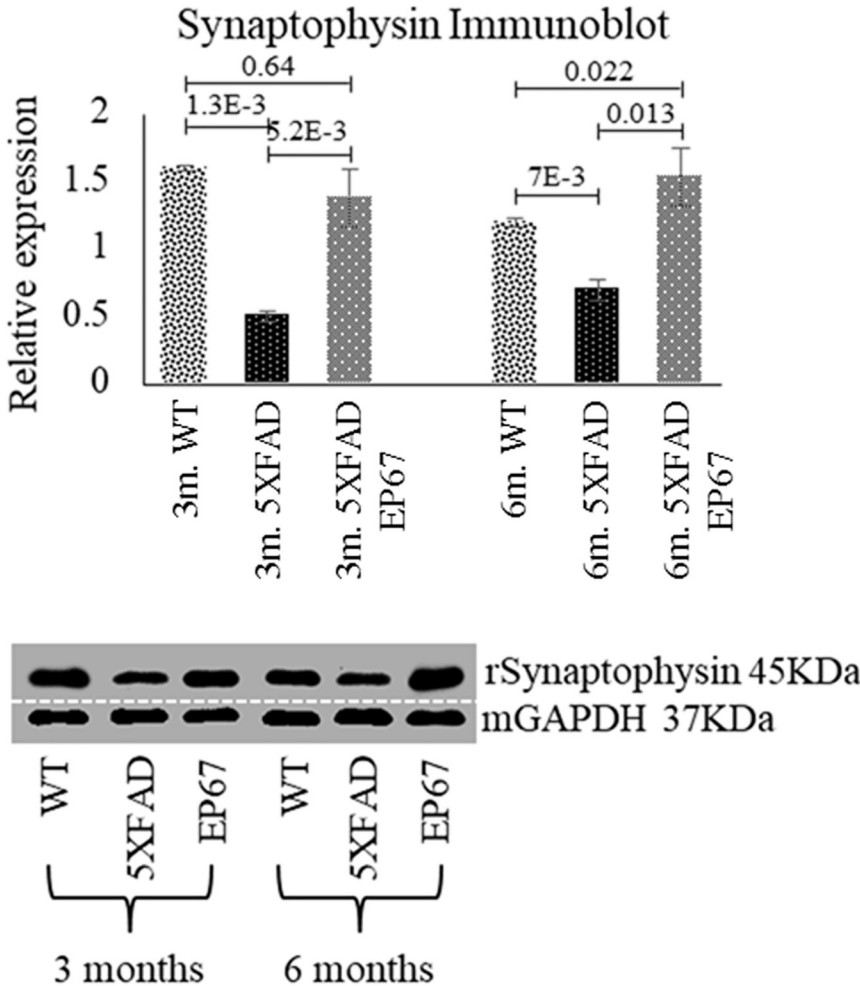

**Fig 6. Synaptophysin immunoblot.** Immunoblots against synaptophysin were prepared. n = 6/group/age. Mean ± 1SD. (Immunoblot images indicate representative sample runs and were cropped as indicated by the dotted white line).

This increase in phagocytosis was also assessed using RT-QPCR which was used to quantify the transcription of specific genes related to this population. Monocytes/macrophages/microglia, incited by pro-inflammatory cytokines, such as TNF-alpha, secrete the monocyte chemoattractant protein-1 (MCP-1), which in turn is responsible for the recruitment of further monocytes into the brain[25]. Our data indicate 5XFAD animals treated with either a single or four cycles of EP67 exhibit a significant increase in MCP-1 when compared to their untreated counterparts (Fig 12a). The monocyte/macrophage/microglia gene transcripts for LY6C and CCR2 were also used to again evaluate whether there's an increase in phagocytosis. Analysis indicates a tremendous increase in both these gene transcripts as previously indicated by the immunohistochemistry images (Fig 12b and 12c). A marker more specific for microglia was not particularly informative and was not significantly elevated in EP67 treated animals (Fig 12d).

Peripheral neutrophils have also been shown to cluster in AD plaques [26]. The RT-QPCR assay for MIP-2a (CXCL2), a chemotactic agent for polymorphonuclear leukocytes secreted by monocytes/macrophages shows a remarkable increase in the EP67 treated animals (Fig 12e).

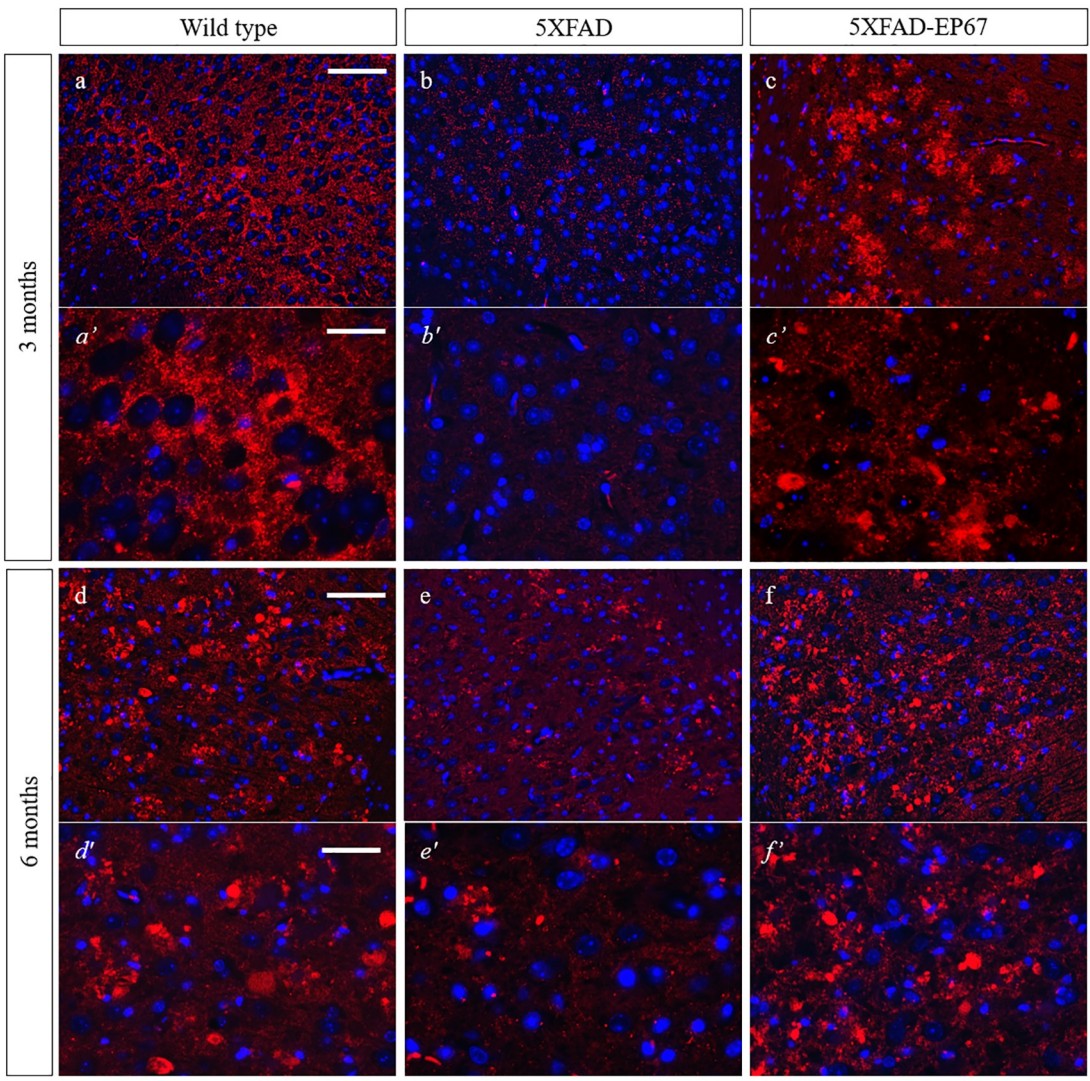

**Fig 7. Synaptophysin expression.** Representative sagittal cortex sections from 3 and 6 month old wild type (a, *a'* & d, *d'*), 5XFAD (b, *b'* & e, *e'*) and 5XFAD EP67 treated (c, *c'* & f, *f'*) mice sections were co-stained with an antibody against Synaptophysin (-red- excitation λ: 358 emission λ:461) and DAPI nuclear staining (-blue- excitation λ: 555 emission λ:580). Scale bars: a-c & d-f 150 µm, *a'- c'* 75 µm and *d'-f'* 48 µm.

Furthermore, immunohistochemical examination using the neutrophil elastase marker, ELANE, in 6-month old control 5XFAD (Fig 13ai–13aiii) and 6-month EP67 treated 5XFAD animals (Fig 13aiv–13avi) did reveal weak ELANE staining, co-localising with Aβ plaques. Using Ly6G for immunoblot, a neutrophil specific marker [27], corroborated the results as with ELANE (Fig 13b).

## Discussion

In a previous study of peripheral nerve amyloidosis, using a double transgenic mouse model of ATTR V30M amyloidosis lacking C1q, we have shown that the absence of the complement C1q results in an increase in amyloid deposition by 60%. This is accompanied by a significant up regulation in apoptotic and cellular stress markers while macrophage recruitment appears to be down regulated[28]. Therefore it appears that complement participates in phagocytosis

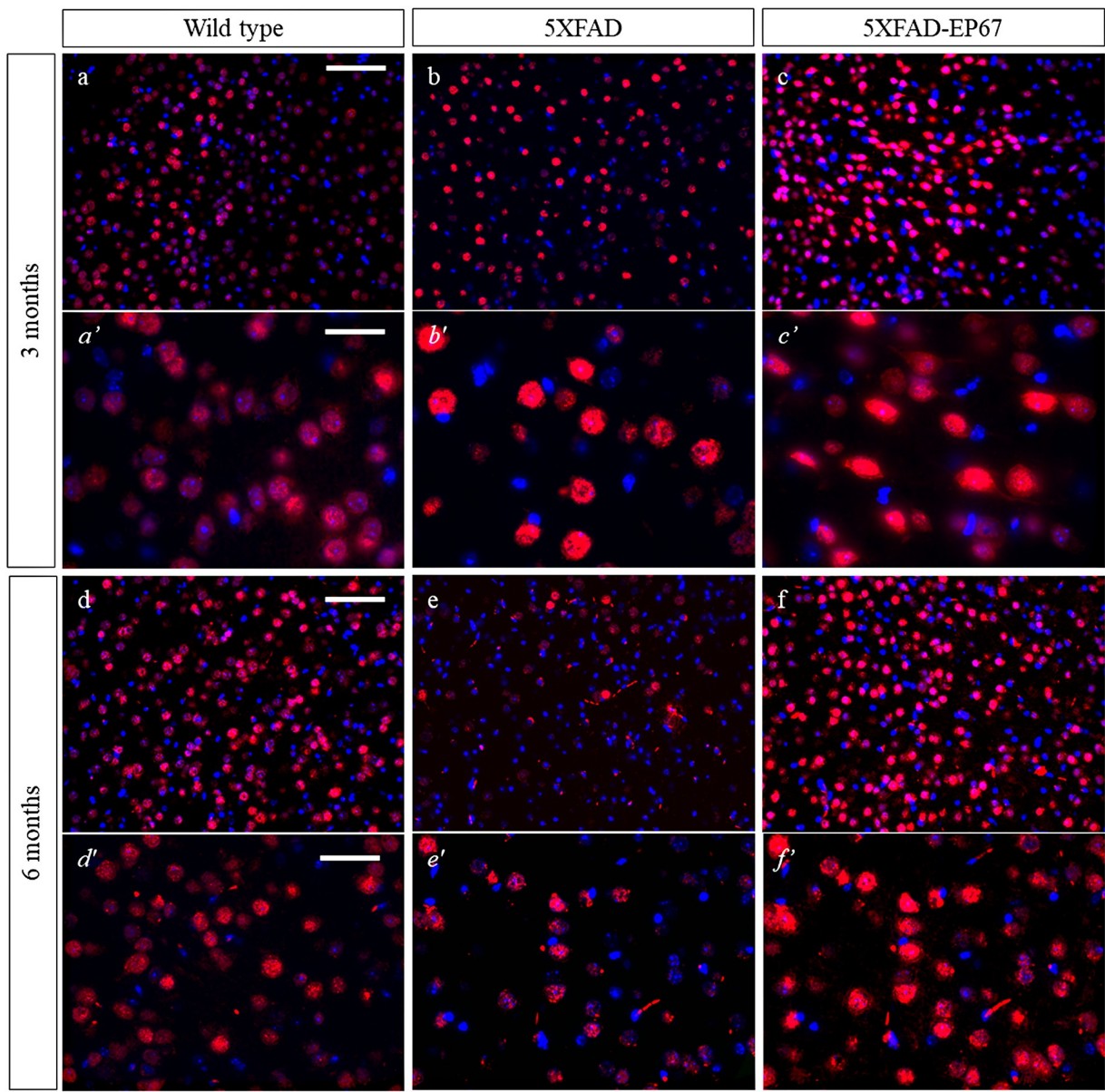

**Fig 8. NeuN expression.** Representative sagittal cortex sections from 3 and 6 month old wild type (a, *a'* & d, *d'*), 5XFAD (b, *b'* & e, *e'*) and 5XFAD EP67 treated (c, *c'* & f, *f'*) mice sections were co-stained with an antibody against NeuN (-red- excitation λ: 358 emission λ:461) and DAPI nuclear staining (-blue- excitation λ: 555 emission λ:580). Scale bars: a-c & d-f 150 μm, *a'- c'* 75 μm and *d'-f'* 48 μm.

of amyloid [28]. According to the amyloid hypothesis of AD the deposition of prefibrillar and fibrillar Aβ peptides sets off pathogenic cascades of neuroinflammation and neurodegeneration that lead to synaptic and neuronal loss and cognitive decline. Various approaches to reduce amyloid load by reducing production of Aβ peptide (secretase inhibition) or enhance amyloid clearance (anti-Aβ peptide and anti-Aβ plaque antibody mediated clearance) in recent Phase 3 trials have not produced clinically meaningful results despite reduction in Aβ load[29–32]. Interfering with the normal function of secretases and suboptimal timing of Aβ peptide removal have been put forward as possible explanations. Timely activation of the innate immune mechanisms, as has been previously been shown in animal model of ATTRMet30

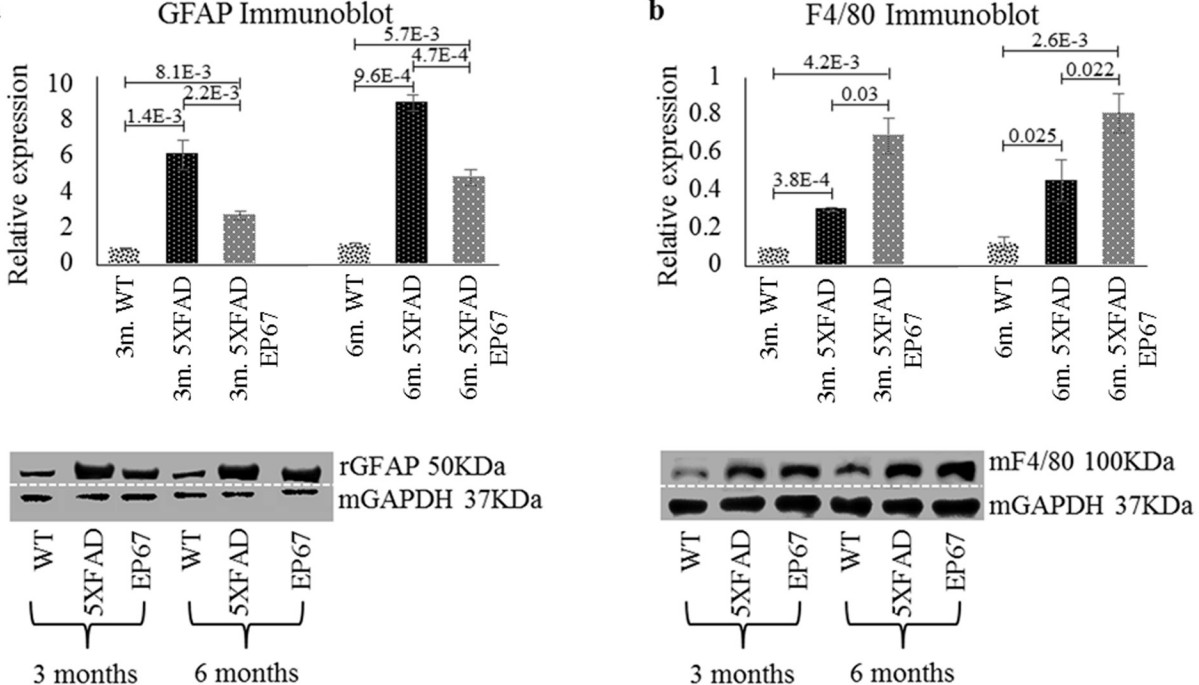

**Fig 9. Astrocytes and macrophages.** (a) Immunoblots against GFAP were prepared. n = 6/group/age. Mean ± 1SD. (b) Immunoblots against F4/80 were also carried out. n = 6/group/age. Mean ± 1SD. (d) (Immunoblot images indicate representative sample runs and were cropped as indicated by the dotted white line).

neuropathy, and presently with 5XFAD mouse model of AD offers an alternative therapeutic avenue[33].

EP67, in the drinking water of 5XFAD mice decreases both non-fibrillar and fibrillar Aβ, preserves neurons and synapses and protects against cognitive impairment. It is becoming increasingly recognized that timely removal of both non-fibrillar and fibrillar Aβ is essential in treating AD [32]. EP67 treated 5XFAD mice exhibited increased phagocytosis and reduced levels of astrocytosis. Aβ peptides have previously been shown to be neurotoxic and induce astrocytosis in vitro[21] as well as trigger changes in astrocyte glutamate uptake and metabolism [34]. In general activated astrocytes are heavily implicated in the inflammatory response observed in AD through the secretion of cytokines and proinflammatory factors [35].

Microglia, the CNS resident macrophages, have been shown to be activated in both AD brains and transgenic disease mouse models, associate with Aβ plaques and contribute to neuroinflammation. In the absence of a stimulus, microglia remain in a deactivated state while secreting neurotrophic and anti-inflammatory factors. In AD brains however, activated microglia and astrocytes flock around amyloid plaques and secrete several inflammatory molecules and cytokines [36, 37]. Chronically activated microglia have been proposed to have a limited capacity of effective phagocytosis of Aβ fragments [9, 38–41]. There is also evidence that in animal models of AD mononuclear phagocytes may be recruited from the periphery to enter the brain and either differentiate into microglia or remain as macrophages [42]

Our data show that monocyte/macrophage/microglia specific markers increase following treatment with the C5a receptor agonist which would indicate that the decrease in both fibrillar and non-fibrillar Aβ may be the result of increased phagocytosis. The fact that neutrophils have also been observed to co-localise with amyloid plaques indicates that there may be a

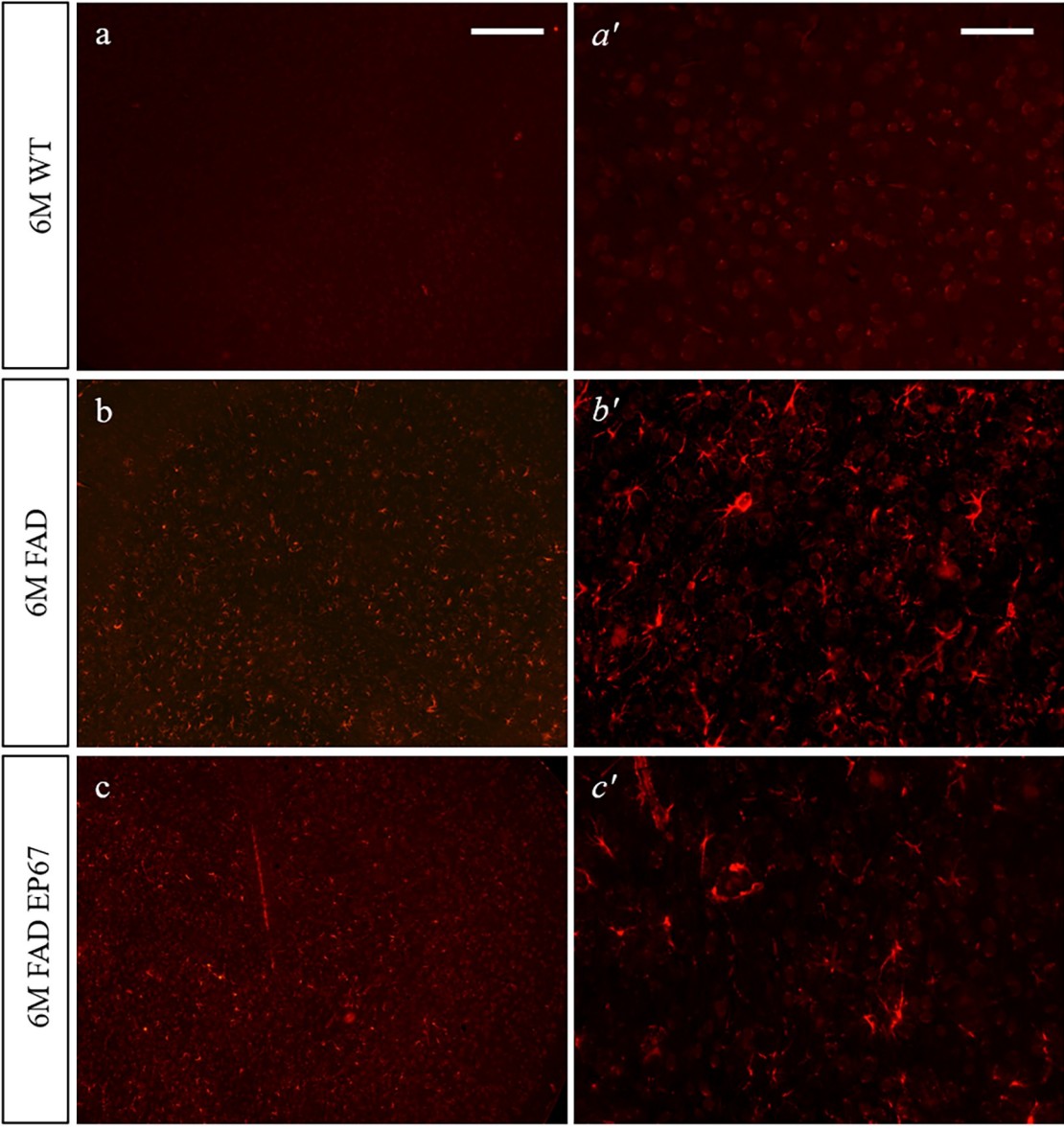

**Fig 10. GFAP expression.** Representative sagittal cortex sections from 6 month old wild type (a & *a'*), 5XFAD (b & *b'*) and 5XFAD EP67 treated (c & *c'*) mice sections were co-stained with an antibody against GFAP (-red- excitation λ: 358 emission λ:461). Scale bars: a-c 150 μm, *a'*- *c'* 75 μm.

contribution from that population in breaking up the amyloid under the influence of EP67. A neutrophil/monocyte marker (Ly6C/G) has been found on cells migrating towards Aβ plaques in an AD mouse model [26]. However it should also be noted that the notion that peripheral phagocytes infiltrate the brain and assist in amyloid clearance has not been convincingly shown in human material. "Patrolling" monocytes have been shown to monitor blood vessels through the effects of LFA-integrin, a molecule which has also been implicated in both neutrophil extravasation into the CNS as well as intraparenchymal motility [43, 44]. Peripheral monocytes which bear many similar markers to resident microglia but found to be morphologically and functionally distinct have also been extensively shown to be recruited into the brain [45, 46]. However, their role as to whether they will assist in amyloid clearance or

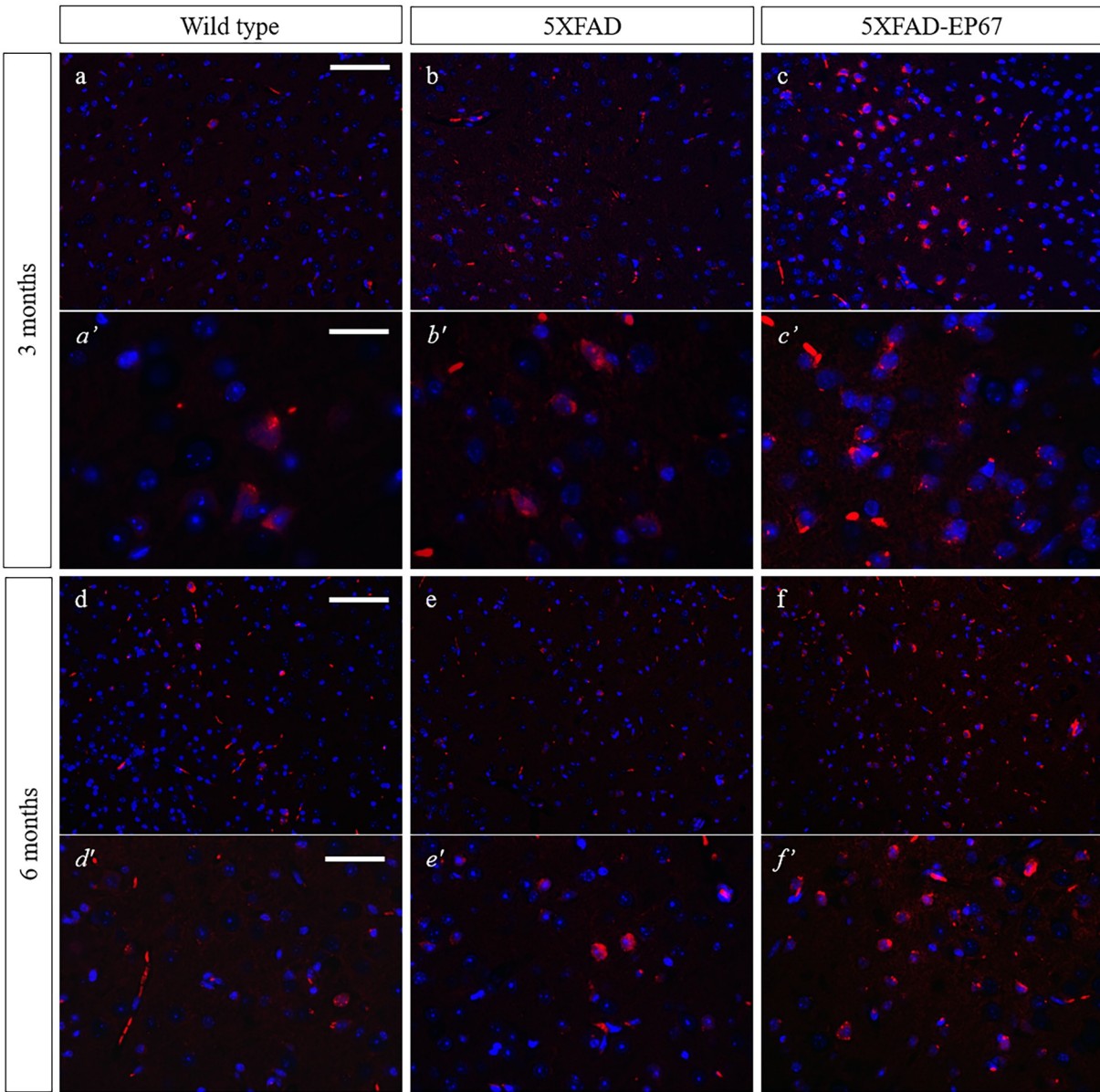

**Fig 11. F4/80 expression.** Representative sagittal cortex sections from 3 and 6 month old wild type (a, *a'* & d, *d'*), 5XFAD (b, *b'* & e, *e'*) and 5XFAD EP67 treated (c, *c'* & f, *f'*) mice sections were co-stained with an antibody against F4/80 (-red- excitation λ: 358 emission λ:461) and DAPI nuclear staining (-blue- excitation λ: 555 emission λ:580). Scale bars: a-c & d-f 150 μm, *a'- c'* 75 μm and *d'-f'* 48 μm.

exacerbate the disease's pathology needs to be settled. A drawback in our study is the fact that we have not directly measured the distribution of EP67 and in particular whether it penetrates into the brain activating phagocytes in situ or whether it acts in the periphery to activate phagocytes which then penetrate the blood brain barrier in response to Aβ amyloid. We are not aware of any studies documenting the distribution of EP67 in the central nervous system. It is likely that EP67 must be absorbed in the systemic circulation for it to have an effect in brain.

There is evidence that microglia activation may have a detrimental role in AD since micro-glia driven neuroinflammation may exacerbate amyloid deposition. PMX205, a C5aR

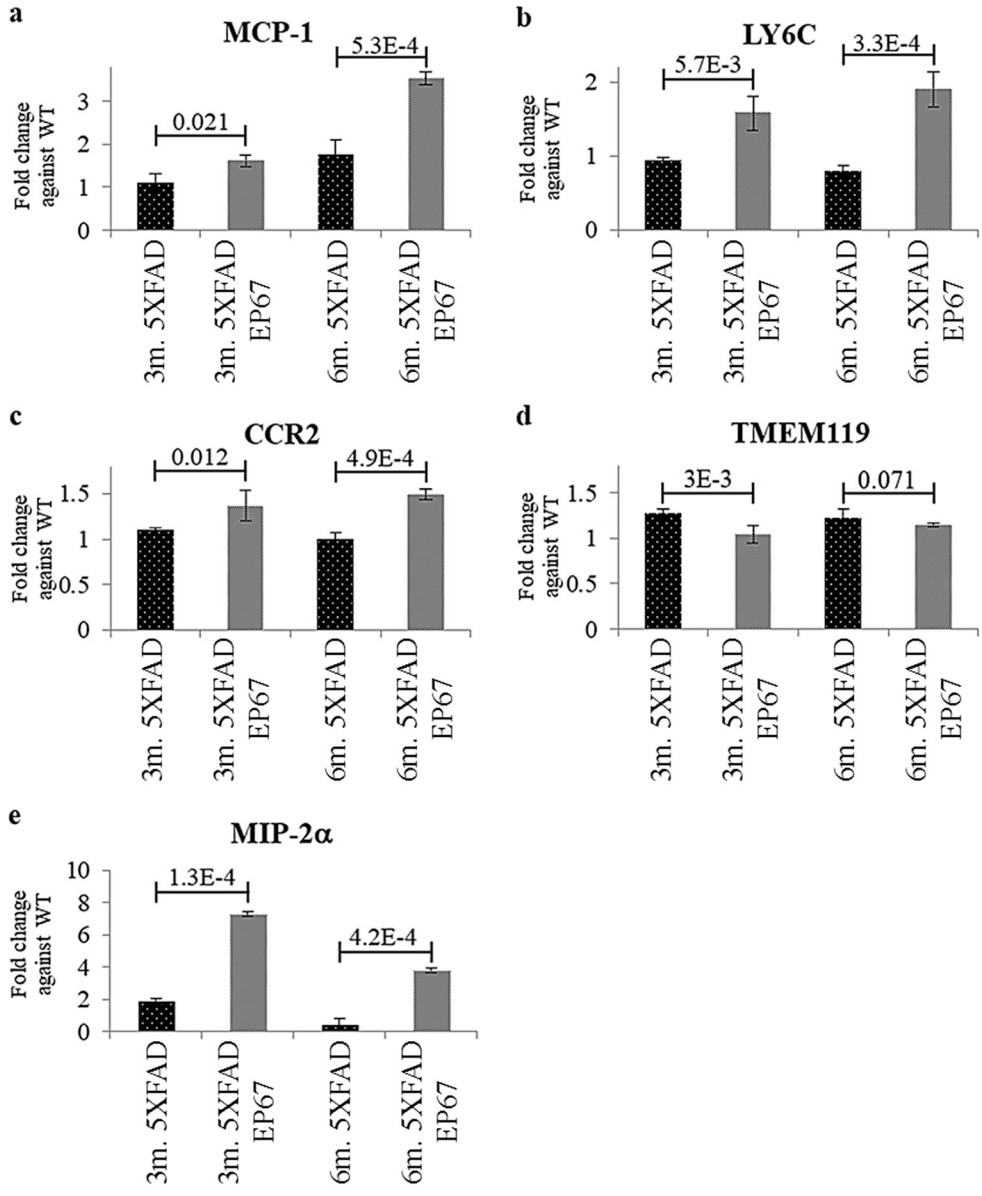

**Fig 12. Phagocytes gene expression levels in the brain.** The levels of gene transcripts were measured through RT-QPCR. (a) MCP-1 a monocyte chemoattractant was found to be significantly increased in animals receiving the EP67 treatment, MIP- 2a (b) the leukocyte chemoattractant appears to be massively overexpressed following treatment with EP67. The expression of the microglia specific marker TMEM119 (c) remains unchanged amongst all groups. CCR2 (d) and LY6C (e), both found to be expressed in monocytes also appear to be significantly increased in animals receiving the EP67 treatment. Comparative CT Method (ΔΔCt) against relevant wild type samples, n = 5/group/age.

antagonist, reduces amyloid deposition, astrocytosis and the microglia response [47]. In addition, C5a has been shown to be directly toxic to neurons [48]. These results appear to contradict our own findings. We believe that the explanation lies in the fact that we have administered the C5aR agonist in an intermittent fashion stimulating cycles of increased phagocytosis, removing amyloid and at the same time avoiding prolonged C5aR stimulation that may lead to continuous microglia/astrocyte stimulation and intense neuroinflammation.

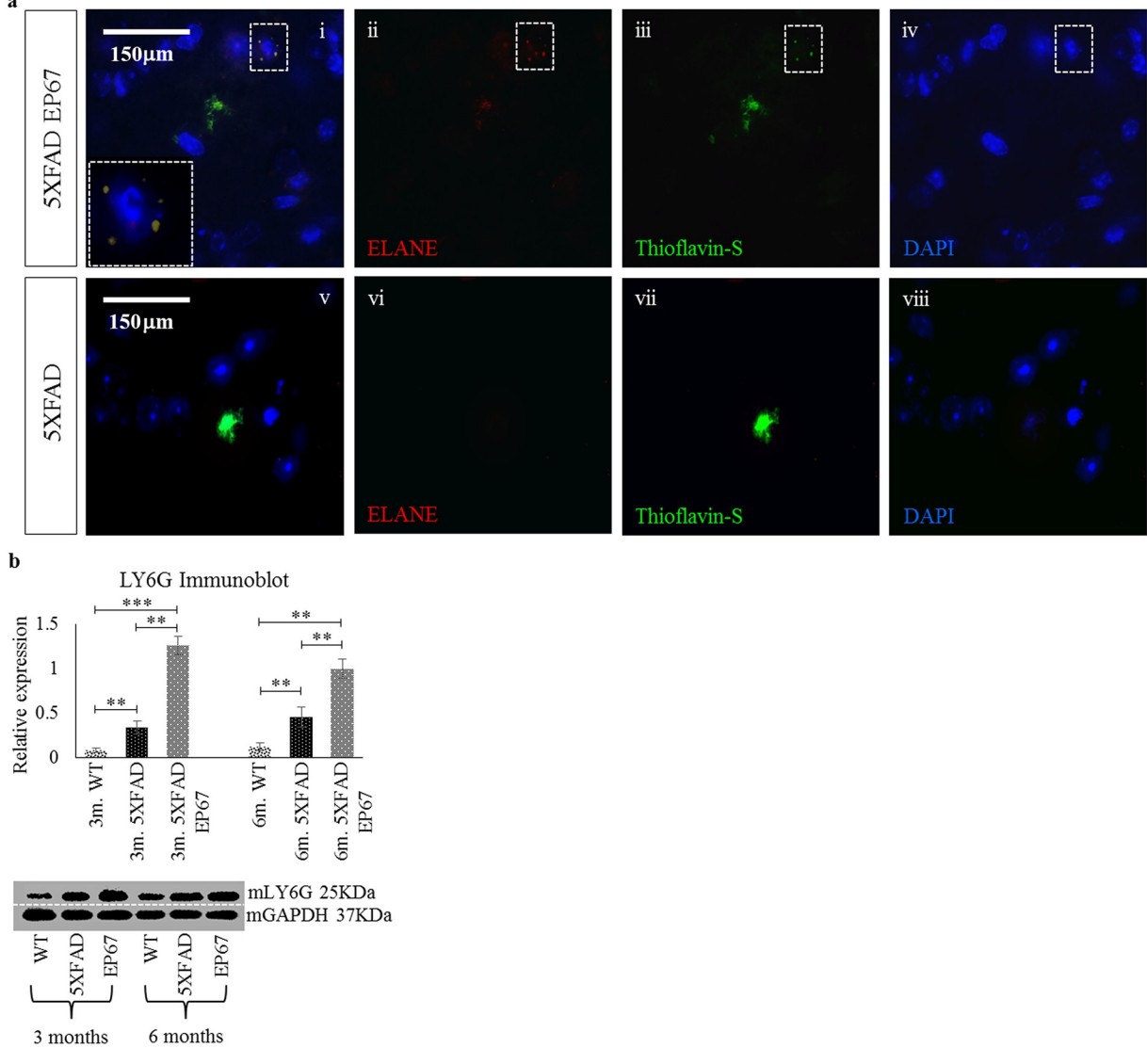

**Fig 13. Neutrophils in the brain.** Representative sagittal cortex sections from 6 month old 5XFAD (i-iii) and 5XFAD EP67 treated (iv-vi) mice were co-stained with Thioflavin-S (-green- excitation λ: 430 emission λ: 550), an antibody against the neutrophil marker Neutrophil Elastase (ELANE) (-red- excitation λ: 555 emission λ:580) and DAPI (-blue- excitation λ: 350 emission λ:470). Immunoblots against another neutrophil marker LY6G were also prepared. n = 6/group/age. Mean ± 1SD. Scale bar: 150 μm. (Immunoblot images indicate representative sample runs and were cropped as indicated by the dotted white line).

Thus removing 'chunks' of amyloid deposits intermittently but avoiding prolonged phagocytic activation and thus driving neuroinflammation may have a net beneficial effect.

Even though we have not directly shown that EP67 penetrates into the brain to activate local phagocytes or whether systemic EP67 activates intravascular monocytes which then penetrate into the brain parenchyma. It is very likely that both phenomena may be occurring. It is becoming apparent that the role of complement in the pathogenesis of AD is particularly complex and probably differs at different stages of the disease. We provide preliminary evidence that timely manipulation of complement perhaps in the presymptomatic phase of Alzheimer disease may be worth pursuing.

## Supporting information

**S1 Fig. Immunoblots.** Raw data from immunoblot experiments.
(TIF)

**S2 Fig. Immunoassays.** Raw data from Aβ42 and Aβ40 immunoassay kits.
(TIF)

**S3 Fig. Y-maze.** Raw data obtained during Y-maze spontaneous alternation test.
(TIF)

**S4 Fig. RT-QPCR.** Raw data from the various real-time PCR targets.
(TIF)

## Author Contributions

**Conceptualization:** Elena Panayiotou, Theodoros Kyriakides.

**Data curation:** Elena Panayiotou.

**Formal analysis:** Elena Panayiotou, Eleni Fella, Petroula Gerasimou, Theodoros Kyriakides.

**Investigation:** Elena Panayiotou, Eleni Fella, Savanna Andreou, Revekka Papacharalambous, Petroula Gerasimou, Stella Angeli, Ioanna Kousiappa.

**Methodology:** Elena Panayiotou, Paul Costeas.

**Project administration:** Theodoros Kyriakides.

**Resources:** Theodoros Kyriakides.

**Supervision:** Paul Costeas, Savvas Papacostas.

**Writing – original draft:** Elena Panayiotou.

**Writing – review & editing:** Theodoros Kyriakides.

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
