## [Decision Letter · Decision Letter 0]

20 Aug 2019

PONE-D-19-19628

C5aR agonist enhances phagocytosis of fibrillar and non-fibrillar Aβ amyloid and preserves memory in a mouse model of Familial Alzheimer’s disease

PLOS ONE

Dear Professor Kyriakides,

Thank you for submitting your manuscript to PLOS ONE. After careful consideration, we feel that it has merit but does not fully meet PLOS ONE’s publication criteria as it currently stands. Therefore, we invite you to submit a revised version of the manuscript that addresses the points raised during the review process.

We would appreciate receiving your revised manuscript by Oct 04 2019 11:59PM. To enhance the reproducibility of your results, we recommend that if applicable you deposit your laboratory protocols in protocols.io, where a protocol can be assigned its own identifier (DOI) such that it can be cited independently in the future. For instructions see: http://journals.plos.org/plosone/s/submission-guidelines#loc-laboratory-protocols

We look forward to receiving your revised manuscript.

Kind regards,

Wataru Araki

Academic Editor

PLOS ONE

Journal Requirements:

2. We note that you have indicated that Avertin was used to sacrifice animals in your study. We would be grateful if you could clarify how death was confirmed in these animals, following administration of Avertin in the dose described. If animals were decapitated or other secondary methods were used to ensure death please include this information in your Methods section.

Please also state whether your ethics committee specifically approved the use of this compound.

Please also provide the supplier of Avertin, or describe how it was synthesised.

3. We also note that you have stated that "All animal experiments were carried out in accordance to the 86/609/EEC Directive (Cyprus Veterinary Services project license approving the experimental and sacrifice protocol CY/EXP/P.L6/2010).".

We would be grateful if you could clarify in your Ethics Statement and Methods section whether the study was approved by an ethics committee. If Cyprus Veterinary Services served as an ethics committee please include this information in your Methods section.

"The authors E.P. and T.K. have filed a PCT application on the use of the modified C5aR agonist EP67 in cerebral and peripheral amyloidoses (PCT/EP2018/053362)."

Additional Editor Comments:

Both reviewers raise some methodological concerns, as described in their comments.  I recommend to make an appropriate revision to address these criticisms carefully. Especially, it is necessary to consider seriously about the validity of some immunohistochemical analyses such as synaptophisin immunostaining. In addition, please provide the method of immunohistochemistry more in detail. Also please recheck the data of Fig 3c, in which very large reductions are observed in 5XFAD, compared with WT.

Reviewers' comments:

Reviewer's Responses to Questions

**Comments to the Author**

1. Is the manuscript technically sound, and do the data support the conclusions?

Reviewer #1: Yes

Reviewer #2: Partly

2. Has the statistical analysis been performed appropriately and rigorously? 

Reviewer #1: Yes

Reviewer #2: Yes

3. Have the authors made all data underlying the findings in their manuscript fully available?

Reviewer #1: No

Reviewer #2: Yes

4. Is the manuscript presented in an intelligible fashion and written in standard English?

Reviewer #1: Yes

Reviewer #2: Yes

5. Review Comments to the Author

Reviewer #1: Pannayiotou et al, PLOS One

This is a relatively straightforward paper demonstrating amyloid reducing effects of a Complement C5a receptor agonist peptide. In general the paper is written clearly, and the results, as shown, appear to support the hypothesis. However, there are some methodological concerns that the authors should try to address. Also, some of the supplemental data should be in the manuscript proper. This referee was not able to access the supplemental material during the review (a frustrating experience), some of which is essential to the arguments.

Issues the authors need to consider addressing in revising the manuscript.

1. The agent used to treat the mice is a 9 amino acid peptide. They administer it orally in the drinking water for a week once each month. Most peptides do not survive transit through the GI tract. There is no demonstration that the drug gets into either plasma or the central nervous system. Do the authors have any pharmacokinetic data they can supply or at least refer to that demonstrates CNS penetration by this relatively hydrophilic peptide? Why did they not provide the agent continuously?

2. The authors claim they modified the peptide such that it activates the macrophage receptor, but not the neutrophil receptor. Are these two receptors different genes, or splice variants? How can this be achieved?

3. The sample size is specified as 6 per group. They state that one hemisphere was fixed for histological processing and the other frozen. They then indicate that the Aβ measurement was performed on the entire hemisphere (line 186). The kits they used specify the use of guanidinium buffer, which would dissolve both fibrillar and soluble Aβ, yet they claim to only measure non-fibrillar Aβ. Was there a centrifugation step?

4. The authors then indicate that they used a RIPA buffer (line 199) to perform immunoblots. Where did the other tissue come from (unless they did not follow the Aβ kit instructions). They further indicate extracting total RNA (line 241). Where did the tissue for this come from? There needs to be some further methodological detail to indicate how they measured these multiple markers from the same tissue. Or perhaps, they did not make the measurements on the same tissue but had either dissected brain regions, more mice, or perhaps reduced the numbers of mice for each measurement.

5. The authors need to find a reasonable convention for specifying the Aβ peptide and stick with it. They variably refer to it as Amyloid β, as αβ, A4, or Aβ.

6. For immunohistochemistry, the authors need to specify the number of sections they analyzed for each stain and the distribution of the sections throughout the tissue. Paraffin sections are often sampling from a limited portion of the region and not very representative overall.

7. Is there evidence that C5a increases the expression of its receptor? Typically, agonists decrease the cognate receptor. As this is the only evidence the authors offer of CNS penetration, it would be more convincing if this was a well known phenomenon.

8 Line 288 seems a bit odd. First, most anti-Aβ antibodies list the aa in the Aβ sequence. It would appear that the aa listed (672-714) are from the amyloid precursor protein sequence, not the Amyloid A4 sequence (which is the same as Aβ). Second, it is also unclear what is meant by A4 precursors

9. The reductions in NeuN staining in the 5xFAD line of 70-80% are beyond any reductions reported previously. These mice have a modest reduction in neuron number in layer V of the anterior cerebral cortex. It is important that the authors specify what regions they are imaging and the number of measurements made per mouse if these data are to be believed. Further the images in the pdf are so dark that this referee is not able to evaluate what is being stained.

10. Figure S1 and S10 should probably be in the manuscript (although this referee was not able to view them). The results presented in S10 at least are integral to the overall interpretation of the mechanism by which this agent appears to prevent amyloid induced changes in the mice.

In summary, this is a manuscript which examines a novel agent for amyloid reducing effects in a mouse model of amyloid deposition (its not really FAD without tau pathology and brain atrophy). There are some methodological issues which need to be addressed. It also should be the case that evidence that the agents does gain access to both plasma and brain should be at least referenced. Given that we have close to 500 manipulations that reduce amyloid in APP mice, it is unclear how much of an advance this is. However, the reported effect sizes are substantial and the approach is relatively novel.

Reviewer #2: In this study, Panayiotou et al., authors demonstrated that the reduction of amyloid-β (Aβ) accumulation in the brains and improvement of cognitive impairment in a model mouse of AD (5XFAD) by the intermittent administration of a modified C5a receptor agonist EP67. Authors also suggested that this therapeutic effects of EP67 are exerted by the preservation of synaptic integrity following the promotion of Aβ clearance by microglia and infiltrated myeloid cells such as monocytes/macrophages but not astrocytes and neutrophils.

This manuscript suggests very important information and evidences for the development of a novel immunotherapeutic strategy against AD: the modulation of brain immunity through the regulation of compliment pathways, especially using a modified C5a receptor agonist, would be the attractive strategy for treating AD.

Text is well written, and discussions are quite fair and reasonable. However, the problem is the poor quality of immunohistochemistry. Except for figure 1a, immunohistochemical data including supplemental figures should be replaced to those of more good quality. Alternatively, it would be possible that authors delete the immunohistochemical analysis in this manuscript as the future study.

For example, staining of Synaptophysin and beta tublin III should be laminar but not dots nor like postsynaptic. In the NeuN staining, the nuclei of neurons are too big. GFAP and F4/80 signals should be detected over a wide area even in wild type mice, Iba1 and CD68 staining should indicate microglia but not neuron like structure. DAPI staining in figure S10 also shows too big nuclei.

Minor comment;

Please define and unify the short form of amyloid-β to ‘αβ’ or ‘Aβ’.

6. PLOS authors have the option to publish the peer review history of their article (what does this mean?). If published, this will include your full peer review and any attached files.

Reviewer #1: No

Reviewer #2: No

---

## [Author Response · Author response to Decision Letter 0]

8 Oct 2019

EDITOR’S COMMENTS

 Response: Changes have been made to adhere to the journal’

2. We note that you have indicated that Avertin was used to sacrifice animals in your study. We would be grateful if you could clarify how death was confirmed in these animals, following administration of Avertin in the dose described. If animals were decapitated or other secondary methods were used to ensure death please include this information in your Methods section.

Please also state whether your ethics committee specifically approved the use of this compound.

Please also provide the supplier of Avertin, or describe how it was synthesised.

Response: Further details have been added in the manuscript in the methods section, lines 156-166.

3. We also note that you have stated that "All animal experiments were carried out in accordance to the 86/609/EEC Directive (Cyprus Veterinary Services project license approving the experimental and sacrifice protocol CY/EXP/P.L6/2010).".

We would be grateful if you could clarify in your Ethics Statement and Methods section whether the study was approved by an ethics committee. If Cyprus Veterinary Services served as an ethics committee please include this information in your Methods section.

Response: The protocol was approved by the Cyprus Veterinary Services, including the sacrifice method, number of animals euthanized and the method of retrieving tissues. This was clarified in the methods section as stated above.

"The authors E.P. and T.K. have filed a PCT application on the use of the modified C5aR agonist EP67 in cerebral and peripheral amyloidoses (PCT/EP2018/053362)."

Response: We confirm that the patent filing does not alter our adherence to all PLOS ONE policies, this was also included in the “Competing interests” section of the manuscript. 

5. Both reviewers raise some methodological concerns, as described in their comments. I recommend to make an appropriate revision to address these criticisms carefully. Especially, it is necessary to consider seriously about the validity of some immunohistochemical analyses such as synaptophisin immunostaining. In addition, please provide the method of immunohistochemistry more in detail. Also please recheck the data of Fig 3c, in which very large reductions are observed in 5XFAD, compared with WT.

Response: We have provided greater detail in the immunohistochemistry protocol and have reviewed our data and figures. New images of the immunohistochemistry experiments have been provided. We should also note that the quantification was only carried out through immunoblotting and immunoassays, while immunohistochemistry was only utilized as a means to confirm/visualize the trend. 

REVIEWER #1 COMMENTS

1. The agent used to treat the mice is a 9 amino acid peptide. They administer it orally in the drinking water for a week once each month. Most peptides do not survive transit through the GI tract. There is no demonstration that the drug gets into either plasma or the central nervous system. Do the authors have any pharmacokinetic data they can supply or at least refer to that demonstrates CNS penetration by this relatively hydrophilic peptide? Why did they not provide the agent continuously?

Response: While we did not specifically investigate for the presence of the peptide in the brain and we could not find any existing literature pertaining to this action, this does not negate the net result. The significant increase of the CD88 receptor (C5aR1) in the brain (Supplementary Figure 2) demonstrates activation of the C5a-C5aR1 axis in the brain but does not prove that this happened locally that was not imported from the periphery. It is likely that both phenomena occur (Fig S3). We do have plans to investigate in depth the pharmacokinetic properties of the EP67 agonist and whether it can or not cross the blood brain barrier. EP67 was not administered continuously to avoid overstimulation of phagocytes and thus exacerbate neuroinflammation with a detrimental effect on amyloidogenesis. 

2. The authors claim they modified the peptide such that it activates the macrophage receptor, but not the neutrophil receptor. Are these two receptors different genes, or splice variants? How can this be achieved?

Response: EP67 is a conformationally biased, response-selective analogue of the biologically active C-terminal region of human complement component C5a 65-74 (ISHKDMQLGR). The agonist was generated by substituting specific residues in C5a and attaching a methyl group to the nitrogen atom on the amide bond between proline and leucine. These structural modifications restrict and extend the backbone’s conformation creating biased topochemical features that allow for a conformational distinction between C5a-like inflammatory and immune stimulatory activity. These features therefore allow EP67 to interact with C5a receptors expressed on antigen presenting cells but is devoid of C5a-like neutrophil and neutropenic abilities (Morgan et al., “Enhancement of in vivo and in vitro immune functions by a conformationally biased, response-selective aagonist of human C5a: Implications for a novel adjuvant in vaccine design.” 2009)..

3. The sample size is specified as 6 per group. They state that one hemisphere was fixed for histological processing and the other frozen. They then indicate that the Aβ measurement was performed on the entire hemisphere (line 186). The kits they used specify the use of guanidinium buffer, which would dissolve both fibrillar and soluble Aβ, yet they claim to only measure non-fibrillar Aβ. Was there a centrifugation step?

Response: Since samples were to be used for western blotting we first pulverized the brain hemisphere with a medical scalpel and homogenized and sonicated the tissue further with PBS supplemented with the protease inhibitor cocktail (approximately 50�L) which resulted in a thick paste consistency. We then used the 100mg of this tissue preparation to carry out sample preparation for ELISA with the guanidine-HCL buffer as stated in the manufacturer’s protocol. The same tissue was also used for western blotting where the proteins were then extracted with RIPA buffer. This has been clarified in the materials and methods section (Lines 170-174). Below is a schematic of which tissue sections were used for which method. Regarding the species measured through the Elisa kits used we had previously contacted the manufacturing company with the same question and were told that the kit can only detect the soluble monomers due to the recognition site of the detection antibody since the C-terminal epitope would not be exposed in aggregated ��. At any rate this is a comparative study between treated and untreated 5XFAD animals.

4. The authors then indicate that they used a RIPA buffer (line 199) to perform immunoblots. Where did the other tissue come from (unless they did not follow the Aβ kit instructions). They further indicate extracting total RNA (line 241). Where did the tissue for this come from? There needs to be some further methodological detail to indicate how they measured these multiple markers from the same tissue. Or perhaps, they did not make the measurements on the same tissue but had either dissected brain regions, more mice, or perhaps reduced the numbers of mice for each measurement.

Response: Sections from the paraffin embedded 4% fixed hemisphere were used for RT-QPCR. This has also been specified in the methods section. Also please refer to lines 170-174, 178. 

5. The authors need to find a reasonable convention for specifying the Aβ peptide and stick with it. They variably refer to it as Amyloid β, as αβ, A4, or Aβ.

Response: Aβ will be used throughout the manuscript, this has been amended accordingly. 

6. For immunohistochemistry, the authors need to specify the number of sections they analyzed for each stain and the distribution of the sections throughout the tissue. Paraffin sections are often sampling from a limited portion of the region and not very representative overall.

Response: Immunohistochemistry was only carried out in order to visualize effects and was not used in a quantitative means, as this was provided by the immunoblotting, immunoassays and RT-QPCR. Fibrillar Aβ quantification with Thioflavin-S was the only immunohistochemistry quantified (please refer to). The conclusions of the current study are mainly based on immunoblots and ELISA while immunohistochemistry sections were used to provide a visual confirmation of the findings. Lines 235-244.

7. Is there evidence that C5a increases the expression of its receptor? Typically, agonists decrease the cognate receptor. As this is the only evidence the authors offer of CNS penetration, it would be more convincing if this was a well known phenomenon.

Response: We did observe and record a significant increase of the CD88 receptor (C5aR1) in the brain (Fig 3) as stated earlier which indicates that EP67 does increase the expression of its receptor in the brain not necessarily on individual cells but due to the increased recruitment of cells bearing the receptors. 

8. Line 288 seems a bit odd. First, most anti-Aβ antibodies list the aa in the Aβ sequence. It would appear that the aa listed (672-714) are from the amyloid precursor protein sequence, not the Amyloid A4 sequence (which is the same as Aβ). Second, it is also unclear what is meant by A4 precursors

Response: This statement has been rephrased, line 304.

9. The reductions in NeuN staining in the 5xFAD line of 70-80% are beyond any reductions reported previously. These mice have a modest reduction in neuron number in layer V of the anterior cerebral cortex. It is important that the authors specify what regions they are imaging and the number of measurements made per mouse if these data are to be believed. Further the images in the pdf are so dark that this referee is not able to evaluate what is being stained.

Response: The images shown were only used for illustration purposes and immunoblots were instead used for quantification, Fig 8 illustrates closer images of the NeuN staining, these have now been taken in greater resolution in order to enable closer observation. The region from which immunochemistry sections are taken are from the posterior cerebral cortex. The immunoblot data are based on homogenates from the whole hemisphere.

10. Figure S1 and S10 should probably be in the manuscript (although this referee was not able to view them). The results presented in S10 at least are integral to the overall interpretation of the mechanism by which this agent appears to prevent amyloid induced changes in the mice.

Response: Supporting figures have now been included in the manuscript.

In summary, this is a manuscript which examines a novel agent for amyloid reducing effects in a mouse model of amyloid deposition (its not really FAD without tau pathology and brain atrophy). There are some methodological issues which need to be addressed. It also should be the case that evidence that the agents does gain access to both plasma and brain should be at least referenced. Given that we have close to 500 manipulations that reduce amyloid in APP mice, it is unclear how much of an advance this is. However, the reported effect sizes are substantial and the approach is relatively novel.

REVIEWER #2 COMMENTS

In this study, Panayiotou et al., authors demonstrated that the reduction of amyloid-β (Aβ) accumulation in the brains and improvement of cognitive impairment in a model mouse of AD (5XFAD) by the intermittent administration of a modified C5a receptor agonist EP67. Authors also suggested that this therapeutic effects of EP67 are exerted by the preservation of synaptic integrity following the promotion of Aβ clearance by microglia and infiltrated myeloid cells such as monocytes/macrophages but not astrocytes and neutrophils.

This manuscript suggests very important information and evidences for the development of a novel immunotherapeutic strategy against AD: the modulation of brain immunity through the regulation of compliment pathways, especially using a modified C5a receptor agonist, would be the attractive strategy for treating AD.

Text is well written, and discussions are quite fair and reasonable. However, the problem is the poor quality of immunohistochemistry. Except for figure 1a, immunohistochemical data including supplemental figures should be replaced to those of more good quality. Alternatively, it would be possible that authors delete the immunohistochemical analysis in this manuscript as the future study.

For example, staining of Synaptophysin and beta tublin III should be laminar but not dots nor like postsynaptic. In the NeuN staining, the nuclei of neurons are too big. GFAP and F4/80 signals should be detected over a wide area even in wild type mice, Iba1 and CD68 staining should indicate microglia but not neuron like structure. DAPI staining in figure S10 also shows too big nuclei.

Minor comment;

Please define and unify the short form of amyloid-β to ‘αβ’ or ‘Aβ’.

Response: The immunohistochemistry images have been deleted from the quantification figures and instead different images have been added. In figure S10 (now Fig 13) there was an error with the scale bar which has been rectified.

---

## [Decision Letter · Decision Letter 1]

21 Oct 2019

PONE-D-19-19628R1

C5aR agonist enhances phagocytosis of fibrillar and non-fibrillar Aβ amyloid and preserves memory in a mouse model of Familial Alzheimer’s disease

PLOS ONE

Dear Professor Kyriakides,

Thank you for submitting your manuscript to PLOS ONE. After careful consideration, we feel that it has merit but does not fully meet PLOS ONE’s publication criteria as it currently stands. Therefore, we invite you to submit a revised version of the manuscript that addresses the points raised during the review process.

In revising your manuscript, please respond to all the comments raised by the editor, which are described below.

We would appreciate receiving your revised manuscript by Dec 05 2019 11:59PM. To enhance the reproducibility of your results, we recommend that if applicable you deposit your laboratory protocols in protocols.io, where a protocol can be assigned its own identifier (DOI) such that it can be cited independently in the future. For instructions see: http://journals.plos.org/plosone/s/submission-guidelines#loc-laboratory-protocols    

We look forward to receiving your revised manuscript.

Kind regards,

Wataru Araki

Academic Editor

PLOS ONE

Additional Editor Comments (if provided):

In the revised manuscript, the authors addressed most of the comments raised by the reviewers. However, their responses to several comments are not sufficient; especially it is necessary to clarify the following points.

1. Regarding the issue of whether EP67 can get into either the plasma or the brain, the authors need to add more explanations in the discussion part.

2. Regarding the comment No2 of Reviewer#1, the response should be added somewhere in the manuscript.

3. Methods of immunoblotting (page 11, 211-216) are not described clearly. This part should be corrected in such a way that other researchers can reproduce the experiments.

4. The subtitles “Fibrillar Abeta quantification” and “Non-fibrillar Abeta quantification” are not appropriate. The former means the quantification of Thioflavin S-positive Abeta plaques and the latter means the quantification of GuHCl-soluble Abeta40 and Abeta42 by ELISA. As the expressions “Fibrillar Abeta quantification” and “Non-fibrillar Abeta quantification” can be misleading, these should be rephrased throughout in the manuscript.

5. Fig 6B: The immunoblotting data of NeuN appears unnatural. As NeuN is a nuclear protein and is not easily extracted by RIPA buffer, the data does not appear to reflect the actual situation in the brain. It is recommended to delete this data and to leave only the immunohistochemistry data.

Reviewers' comments:

Reviewer's Responses to Questions

**Comments to the Author**

Reviewer #2: All comments have been addressed

2. Is the manuscript technically sound, and do the data support the conclusions?

Reviewer #2: Yes

3. Has the statistical analysis been performed appropriately and rigorously? 

Reviewer #2: Yes

4. Have the authors made all data underlying the findings in their manuscript fully available?

Reviewer #2: Yes

5. Is the manuscript presented in an intelligible fashion and written in standard English?

Reviewer #2: Yes

6. Review Comments to the Author

Reviewer #2: This manuscript has been addresed all my comments and revised properly. Statistical analysis been performed appropriately. Authors has made all data underlying the findings in their manuscript fully available, and the manuscript presented in an intelligible fashion and written in standard English.

7. PLOS authors have the option to publish the peer review history of their article (what does this mean?). If published, this will include your full peer review and any attached files.

Reviewer #2: Yes: Kazuyuki Takata

---

## [Author Response · Author response to Decision Letter 1]

23 Oct 2019

1. Regarding the issue of whether EP67 can get into either the plasma or the brain, the authors need to add more explanations in the discussion part.

Response: This has been added to the manuscript in the discussion section, lines 544-550. 

2. Regarding the comment No2 of Reviewer#1, the response should be added somewhere in the manuscript.

Response: This information has been added, lines 101-110.

3. Methods of immunoblotting (page 11, 211-216) are not described clearly. This part should be corrected in such a way that other researchers can reproduce the experiments.

Response: More details were added in the manuscript, lines 214-226.

4. The subtitles “Fibrillar Abeta quantification” and “Non-fibrillar Abeta quantification” are not appropriate. The former means the quantification of Thioflavin S-positive Abeta plaques and the latter means the quantification of GuHCl-soluble Abeta40 and Abeta42 by ELISA. As the expressions “Fibrillar Abeta quantification” and “Non-fibrillar Abeta quantification” can be misleading, these should be rephrased throughout in the manuscript.

Response: The subtitles were changed accordingly, lines 175 and 195. 

5. Fig 6B: The immunoblotting data of NeuN appears unnatural. As NeuN is a nuclear protein and is not easily extracted by RIPA buffer, the data does not appear to reflect the actual situation in the brain. It is recommended to delete this data and to leave only the immunohistochemistry data.

Response: This immunoblot was deleted and modifications were made, lines 373-375, 385-387 and Fig 6.

---

## [Editor Report · Decision Letter 2]

30 Oct 2019

PONE-D-19-19628R2

C5aR agonist enhances phagocytosis of fibrillar and non-fibrillar Aβ amyloid and preserves memory in a mouse model of Familial Alzheimer’s disease

PLOS ONE

Dear Professor Kyriakides,

Thank you for submitting your manuscript to PLOS ONE. After careful consideration, we feel that it has merit but does not fully meet PLOS ONE’s publication criteria as it currently stands. Therefore, we invite you to submit a revised version of the manuscript that addresses the points raised during the review process.

Please revise carefully according to the editor's comments, including the previous ones.

We would appreciate receiving your revised manuscript by Dec 14 2019 11:59PM. To enhance the reproducibility of your results, we recommend that if applicable you deposit your laboratory protocols in protocols.io, where a protocol can be assigned its own identifier (DOI) such that it can be cited independently in the future. For instructions see: http://journals.plos.org/plosone/s/submission-guidelines#loc-laboratory-protocols

We look forward to receiving your revised manuscript.

Kind regards,

Wataru Araki

Academic Editor

PLOS ONE

Additional Editor Comments (if provided):

In the revised manuscript, the authors’ responses are not complete.

The editor noted some points need to be clarified, as follows.

The responses to comment No3 are not complete.

Please correct the following points.

L189 “Fibrillar Abeta quantification”

L195 “soluble Abeta quantification” This means GuHCl-soluble.

L323 “non-fibrillar peptides”

L197-198 “manufacturer’s instructions” This part should be described more in details.

L373-375 “neuronal loss”

Synaptophysin loss indicates synaptic loss, not neuronal loss.

This part should be changed to “synaptic and neuronal loss”.

---

## [Author Response · Author response to Decision Letter 2]

1 Nov 2019

Additional Editor Comments:

1. The responses to comment No3 are not complete.

Response: Further details have been added to the immunoblotting protocol in the manuscript, lines 227-257.

2. Please correct the following points.

• L189 “Fibrillar Abeta quantification”

Response: This heading has been changed, line 191.

• L195 “soluble Abeta quantification” This means GuHCl-soluble.

Response: This has been changed lines 197 and 198. 

• L323 “non-fibrillar peptides”

Response: This has been changed line 343. 

• L197-198 “manufacturer’s instructions” This part should be described more in details.

Response: Further details have been added to the immunoassay protocol in the manuscript, lines 199-208. 

• L373-375 “neuronal loss”

Synaptophysin loss indicates synaptic loss, not neuronal loss.

This part should be changed to “synaptic and neuronal loss”.

Response: This has been changed lines 394 and 395.

---

## [Editor Report · Decision Letter 3]

5 Nov 2019

C5aR agonist enhances phagocytosis of fibrillar and non-fibrillar Aβ amyloid and preserves memory in a mouse model of Familial Alzheimer’s disease

PONE-D-19-19628R3

Dear Dr. Kyriakides,

We are pleased to inform you that your manuscript has been judged scientifically suitable for publication and will be formally accepted for publication once it complies with all outstanding technical requirements.

With kind regards,

Wataru Araki

Academic Editor

PLOS ONE

Additional Editor Comments (optional):

The authors have made an appropriate revision.
---

## [Editor Report · Acceptance letter]

21 Nov 2019

PONE-D-19-19628R3 

C5aR agonist enhances phagocytosis of fibrillar and non-fibrillar Aβ amyloid and preserves memory in a mouse model of Familial Alzheimer’s disease 

Dear Dr. Kyriakides:

I am pleased to inform you that your manuscript has been deemed suitable for publication in PLOS ONE. Congratulations! Your manuscript is now with our production department. 

With kind regards,

on behalf of

Dr. Wataru Araki 

Academic Editor

PLOS ONE